# Technical Complications of Removable Partial Dentures in the Moderately Reduced Dentition: A Systematic Review

**DOI:** 10.3390/dj11020055

**Published:** 2023-02-20

**Authors:** Marie-Theres Dawid, Ovidiu Moldovan, Heike Rudolph, Katharina Kuhn, Ralph G. Luthardt

**Affiliations:** 1Department of Prosthetic Dentistry, Center of Dentistry, Ulm University, Albert-Einstein-Allee 11, 89081 Ulm, Germany; 2Private Practice, Philippine-Welser-Str. 15, 86150 Augsburg, Germany

**Keywords:** RPD, removable partial denture, technical complications, systematic review

## Abstract

The aim of this study was to conduct a systematic literature review with a subsequent meta-analysis on the technical complications and failures of removable partial denture (RPD) therapy in the moderately reduced dentition. A systematic literature search of established medical databases, last updated 06/2022, was conducted. RCTs and prospective and retrospective studies were included that had information on technical complications and failures of RPDs, at least 15 participants, an observation period of at least two years and a drop-out rate of less than 25%. Publications were selected on the title, abstract and full-text level by at least three of the participating authors. The evidence of the included studies was classified using the GRADE system. The bias risk was determined using the RoB2 tool and the ROBINS-I tool. Of 19,592 initial hits, 43 publications were included. Predominantly, retention of the prosthesis, retention loss of anchor crowns (decementations), fractures/repairs of frameworks, denture teeth, veneering or acrylic bases, and a need for relining were reported depending on prosthesis type and observation time. Focusing on technical complications and failures, only very heterogeneous data were found and publications with the highest quality level according to GRADE were scarce. Whenever possible, data on technical complications and failures should be reported separately when referencing the tooth, the prosthesis and the patient for comparability. Prostheses with differing anchorage types should be analyzed in different groups, as the respective complications and failures differ. A precise description of the kinds of complications and failures, as well as of the resulting follow-up treatment measures, should be given.

## 1. Introduction

Removable partial dentures (RPDs) are still the most common form of prosthetic treatment for moderately (four or more teeth) to severely (three to one remaining teeth) reduced dentition. They differ primarily in the type of anchorage elements that connect them to the remaining teeth: clasps, precision attachments, bars or double crowns in different variations. Due to the high design variability, removable prostheses can be fabricated in a range from less invasive, technically relatively simple and inexpensive, to invasive, technically complex and expensive.

The long-term success of the therapy and the corresponding restorations depends on both biological and technical parameters. Two previous systematic literature reviews analyzed survival rates, biological complications, and failures of differently retained prosthesis types [1,2]. The impact of complex, technical manufacturing methods on long-term success is particularly relevant from a socioeconomic point of view. In addition to the initial costs, the follow-up costs due to the susceptibility of RPDs to necessary repairs are also important to consider.

Thus, the aim of the present publication was to evaluate the technical complications and failures of removable partial dentures in moderately reduced dentition through a systematic literature review. If possible, a meta-analysis should be performed; however, we were unable to perform one due to inconsistent parameters.

## 2. Materials and Methods

The PRISMA guideline [3] served as the basis for this systematic review. The completed checklist is also available online.

The main question was developed using the PICO scheme (Table 1).

According to the inclusion criteria, RCTs, prospective and retrospective controlled clinical trials, and single- or multiple-arm longitudinal studies were analyzed for this article. Systematic reviews and meta-analyses were also considered, but only on the basis of the original publications. All kinds of anchoring elements were included. The primary inclusion criterion was evaluable data on time-related technical failure and/or complication rates. Articles with data on failure or survival rates and on technical complication rates were included, as well as articles with data on technical complications only.

Regardless of the Kennedy class, all removable restorations that did not meet the exclusion criteria were included. This is to say, four or more teeth characterize a moderately reduced dentition, while a severely reduced dentition implies only three to one remaining teeth. Restorations for edentulous patients and all fixed restorations were excluded. Studies that investigated both removable and fixed prostheses in different study arms were not primarily excluded.

The exclusion criteria included:The language was neither German nor EnglishThere were three or fewer residual teethThe number of cases or participants was smaller than 15The observation duration of the entire cohort was less than two yearsThe drop-out rate was higher than 25%Studies were on complete dentures, cover dentures (complete telescopic dentures/resilient telescopic dentures), simple acrylic dentures (interim dentures) and implant-supported denturesCase reportsPilot studiesAbstract publicationsStudies without an adequate definition of inclusion or failure criteria (narrative presentation).

The search terms, the search strategy, the databases used, and the time periods were described in two previous publications [1,2].

Over the course of this systematic review, the database search in PubMed was updated on a regular basis and last performed on 29 June 2022. The original search included additional databases (Appendix A). The hand search ended on 15 January 2014. The start dates of the hand search can be found in the Appendix A and include all volumes of the respective journals. Since neither additional electronic databases nor the hand search revealed additional results on a full-text level compared with the PubMed search, these additional sources were not considered for the update searches. A detailed description of the search process can be found in the Appendix A.

The initial selection of retrieved articles was conducted generously (“in doubt, leave it in”) at the title and abstract level by at least two independent reviewers based on the inclusion and exclusion criteria. All potentially relevant publications were available in full text and also analyzed independently by at least two authors. Final decisions on inclusion or exclusion were made during joint consensus conferences (H.R., R.G.L. and O.M.). Included publications of the updated search were tabulated by one author (M-T.D.) and were evaluated and controlled by coauthors (H.R. and K.K.) (Table 2).

The selected publications were classified according to their respective level of evidence and examined with regard to their bias risk. The GRADE system [4] was used for this purpose. The quality of the included studies was then assessed based on the quality assessment criteria of Guyatt et al. [4]. RCTs were initially classified as high quality (+ + + +), and retrospective or prospective studies without control were classified as low (+ +) (Table 3). In cases where there was a risk of bias/limitations, inconsistencies or imprecisions, the study was downgraded by one level. The rationale for the downgrade is set out in Table 2 and Appendix A.

**Table 2 dentistry-11-00055-t002:** Characteristics of the included studies.

Author, Year	Study Type	Observation Time	Kind of RPD	Patients Examined and Included in Evaluation	Control Group	Follow-Up Examinations (Planned)	Further Information
Al-Imam, 2016 [5]	Retrospective	1–5 years	Clasp-retained	65 patients with 83 RDPs	n.a.	1–2 months after treatment and 1–5 years after treatment	Prosthesis design not described; 49 patients with RDPs over 2 years in situ
Au, 2000 [6]	RCT	2 years	Clasp-retained	14 patients with 19 titanium RPDs	16 patients with 28 cobalt-chromium RPDs	1, 6, 12, and 24 months	n.a.
Behr, 2000 [7]	Retrospective	Average observation time for parallel-sided: 4.6 ± 1.6 years (1.2–6.8 years); conical: 5.2 ± 1.3 years (1.8–6.8 years)	Double-crown-retained: parallel-sided and conical	117 patients: 74 patients with parallel-sided crown-retained dentures (251 abutment teeth) and 43 patients with conical crown-retained dentures (160 abutment teeth)	n.a.	Regular follow-up investigations	Only decementations per prosthesis mentioned
Behr, 2009 [8]	Retrospective	1984–2007	Double-crown-retained: conical, parallel-sided with clearance fit and parallel-sided with friction fit	577 patients with 577 RPDs: 62 conical crown-retained RPDs, 315 parallel-sided telescopic crowns with clearance fit and 200 parallel-sided telescopic crowns with friction fit	n.a.	n.a.	Only decementations per prosthesis mentioned
Behr, 2012 [9]	Retrospective	Median follow-up time: 3 years	Clasp-retained	174 patients with 179 RDPs: 169 cobalt-chromium, 3 noble alloy and 2 titanium RDPs	n.a.	Once a year	n.a.
Bergman, 1971 [10]	Retrospective	2 years	Clasp-retained	29 patients with RDPs (no specific number of RDPs)	10 patients without treatment	1–2 weeks after end of treatment, and after 12 and 24 months	n.a.
Bergman, 1977 [11]	Retrospective	6 years	Clasp-retained	28 patients with RDPs (no specific number of RDPs)	10 patients without treatment	1–2 weeks after end of treatment, and after 1, 2, 4, and 6 years (also control examination without recording after 3 and 5 years)	Same patient collective as Bergman, 1971
Bergman, 1982 [12]	Retrospective	10 years	Clasp-retained	27 patients with RDPs (no specific number of RDPs)	10 patients until 6 years after treatment	1–2 weeks after end of treatment, and after 1, 2, 4, 6, and 10 years (also control examination without recording after 3, 5, 7, 8, and 9 years)	Same patient collective as Bergman, 1971
Bergmann, 1996 [13]	Retrospective	73–92 months	Double-crown-retained: conical	18 patients with 18 RDPs (78 abutment teeth)	n.a.	Re-examination I: 9–28 months after treatment; re-examination II: 24–43 months; re-examination III: 48–67 months; re-examination IV: 73–92 months	Same patient collective as Ericson, 1990
Budtz-Jørgensen, 1987 [14]	RCT	2 years	Clasp-retained	26 patients with RDPs	27 patients with distal cantilever bridges	1–2 months after treatment, and 6, 12, 18, and 24 months after treatment	n.a.
Budtz-Jørgensen, 1990 [15]	RCT	5 years	Clasp-retained	26 patients with RDPs	27 patients with distal cantilever bridges	1–2 months after treatment and once a year after treatment	Same patient collective as Budtz-Jørgensen, 1987
Ericson, 1990 [16]	Retrospective	24–43 months	Double-crown-retained: conical	23 patients with 24 RDPs (96 abutment teeth)	n.a.	Re-examination I: 9–28 months after treatment; re-examination II: 24–43 months	n.a.
Hahnel, 2012 [17]	Retrospective	Mean observation time for conical: 8.9 ± 5.2 years; clearance fit: 2.8 ± 2.8 years; friction fit: 3.9 ± 3.8 years	Double-crown-retained: conical, parallel-sided with a clearance fit and parallel-sided with friction fit	575 patients with 575 RPDs (1807 abutment teeth): 61 conical crown-retained RPDs, 315 parallel-sided telescopic crowns with clearance fit and 199 parallel-sided telescopic crowns with friction fit	n.a.	n.a.	Same patient collective as Behr, 2009
Heydecke, 2003 [18]	Retrospective	5 years	Attachment-retained: individual attachment with prefabricated spring lock retention element (FR-Chip)	Mean after 27 months: 47 patients with 55 RDPs; 59 ± 11 months: 34 patients with 40 RDPs	n.a.	6, 12, and 24 months, and 5 years after treatment	All results related to the number of patients after 27 months (lack of differentiation)
Ishida, 2017 [19]	Retrospective	Mean observation period: 38.0 ± 20.3 months	Clasp-retained; double-crown retained: conical, resilient and electroplated	201 patients with 52 double-crown-retained RDPs (144 D-teeth: 92 cast conical crowns, 10 resilient telescopic crowns and 42 electroplated double crowns) and 199 clasp-retained RDPs (399 abutment teeth)	n.a.	Twice a year	Different types of double crowns: all types summarized for the evaluation (lack of differentiation)
Kapur, 1989, Part II [20]	RCT	5 years	Clasp-retained	118 patients with 118 RDPs	114 patients treated with fixed partial dentures with implants	Every 6 months with study examinations after 6, 18, 36, and 60 months after treatment	Only male patients and only mandibular RDPs
Kapur, 1994, Part I [21]	RCT	5 years	Clasp-retained: circumferential design and bar design	59 patients with 59 RDPs with circumferential design	59 patients with 59 RDPs with bar design	16 weeks, and 6, 18, 36, and 60 months after treatment	Only male patients and only mandibular RDPs
Kurosaki, 2021 [22]	Retrospective	Mean observation period: 6.1 ± 1.2 years (5.0–8.2 years)	Clasp-retained	20 patients with RDPs	58 patients with implant-fixed partial dentures and 27 patients with fixed partial dentures	6 years after treatment	Multiple reasons for a complication summarized
Mock, 2005 [23]	Prospective	Mean observation period: 7.4 years; up to 10 years	Double-crown-retained: parallel-sided with friction fit	92 patients with 105 RDPs (299 abutment teeth)	n.a.	First 8 follow-up examinations: every 6 months; from follow-up treatment number 9: once a year	n.a.
Nickenig, 1995 [24]	Retrospective	1980–1992	Double-crown-retained: mostly cylindric double-crown-retained	85 patients with 105 RDPs (402 abutment teeth)	n.a.	n.a.	n.a.
Nisser, 2022 [25]	Retrospective	Mean observation period: 44.9 ± 30.8 months (6.2–120.5 months)	Clasp-retained and attachment-retained	142 patients with 172 RDPs (541 abutment teeth): 142 RDPs with clasps and 30 RDPs with intra- and/or extra-coronal attachments	n.a.	n.a.	Different types of RDPs summarized for the evaluation (lack of differentiation)
Pihlaja, 2015 [26]	Retrospective	Mean observation period: 4.2 years (2.9–5.4 years)	Clasp-retained	17 patients with 17 RDPs (37 abutment teeth)	n.a.	n.a.	Clasp-bearing crowns were made of zirconium dioxide
Rehmann, 2006 [27]	Retrospective	Mean observation period: 5.3 ± 2.9 years	Double-crown-retained: cylindric and cylindric with clasps	554 RDPs (1758 abutment teeth): 524 RDPs with cylindric double crowns and 30 RDPs with cylindric double crowns that are also retained with cast clasps on the molars	n.a.	n.a.	Same patient collective as Wöstmann, 2007; number of patients not specified; different types of RDPs summarized (lack of differentiation)
Schmitt, 2011 [28]	Prospective	5 years	Attachment-retained: extra-coronal attachment with interchangeable plastic inserts that are adjustable with activation screw and spring bolt attachments	Kennedy-class I: 20 RDPs with extra-coronal attachment with interchangeable plastic inserts that are adjustable with activation screw (43 attachments); Kennedy-class II: 8 RDPs with spring bolt attachments (8 attachments)	n.a.	2 weeks, and 1, 2, 3, and 5 years after treatment	n.a.
Scholz, 2010 [29]	Retrospective	61 months (54–72 months)	Double-crown-retained: conical	48 patients with 73 RDPs (248 facings)	n.a.	Once a year	n.a.
Schulte, 1980 [30]	Retrospective	Mean service time: 2.5 years	Others (swing lock-retained)	57 patients with 53 swing lock RDPs	n.a.	n.a.	n.a.
Schwindling, 2014 [31]	Retrospective	Mean observation period: 6.26 ± 2.2 years (1.9–8.9 years)	Double-crown-retained: telescopic, conical and resilient	86 patients with 117 RDPs: 32 telescopic crown-retained RDPs, 51 conical crown-retained RDPs and 34 resilient telescopic crown-retained overdentures	n.a.	One examination after mean observation period 6.26 ± 2.2 years (1.9–8.9 years)	Different types of double crowns: all types summarized for the evaluation (lack of differentiation)
Schwindling, 2017 [32]	RCT	3 years	Double-crown-retained: electroplated	27 patients with 30 electroplated double-crown RDPs with zirconia primary crowns	29 patients with 30 electroplated double-crown RDPs with cast cobalt-chromium primary crowns	17.2 ± 3.3 months in the study group/17.2 ± 2.9 months in the control group; 6, 12, 24, and 36 months after treatment	n.a.
Stegelmann, 2012 [33]	Retrospective	Median observation time: 28 months for clasp-retained RDPs; 49 months for attachment-retained RDPs (4–141 months)	Clasp-retained and attachment-retained	203 patients with 329 RDPS: 135 attachment-retained RDPs and 68 clasp-retained RDPs	n.a.	n.a.	n.a.
Stober, 2012 [34]	RCT	3 years	Double-crown-retained: conical and electroplated	54 patients with 60 RDPs (217 abutment teeth): 30 conical double-crown-retained RDPs	30 electroplated double-crown-retained RDPs	6, 12, 24, and 36 months after treatment	Number of patients per group is missing
Stober, 2015 [35]	RCT	72 months ± 4 weeks	Double-crown-retained: conical and electroplated	54 patients with 60 RDPs (217 abutment teeth): 30 conical double-crown-retained RDPs	30 electroplated double-crown-retained RDPs	6, 12, 24, 36, 48, 60, and 72 months after treatment	Same patient collective as Stober, 2012; number of patients per group is missing
Stober, 2020 [36]	Prospective	2 years	Double-crown-retained	30 patients (157 denture teeth; number of prostheses not specified)	32 patients with 47 complete dentures	4 weeks and 24 months after treatment	No precise description of the kind of double crown
Thomason, 2007 [37]	RCT	5 years	Clasp-retained	30 patients with RDPs	30 patients with cantilever resin-bonded fixed partial dentures	3 months, 1 year and then once a year after treatment	Multiple reasons for a complication summarized
Vanzeveren, Part I, 2003 [38]	Retrospective	RDPs were made in 1983–1994 and re-examined in 1998–2000	Clasp-retained and attachment-retained	254 patients with 292 RDPs (some of them with attachments)	n.a.	Re-examination 1998–2000	Different types of RDPs summarized for the evaluation (lack of differentiation)
Vanzeveren, Part II, 2003 [39]	Retrospective	RDPs were made in 1983–1994 and re-examined in 1998–2000	Clasp-retained and attachment-retained	254 patients with 292 RDPs (some of them with attachments) (804 abutment teeth)	n.a.	Re-examination 1998–2000	Same patient collective as Vanzeveren, part I; different types of RDPs summarized for the evaluation (lack of differentiation)
Vermeulen, 1996 [40]	Retrospective	10 years	Clasps and others (Dolder bar and ball or Dalbo attachments)	748 patients with 703 clasp-retained RDPs and 183 attachment-retained RDPs	n.a.	Every 6 months	n.a.
Wagner, 2000 [41]	Retrospective	10 years	(1) Double-crown-retained: conical and conical with clasps on molars; (2) clasp-retained	65: 43 conical crown-retained RDPs, 6 clasp-retained RPDs, and 16 conical crowns on anterior teeth and clasps on molars combined in a single denture	n.a.	After 10 years	65 + 7 = 72 RDPs if 7 RDPs modified to complete dentures were also considered (only sometimes); different types of RDPs summarized for the evaluation (lack of differentiation)
Wenz, 2001 [42]	Retrospective	4.1 ± 3.6 years (0.5–14.4 years)	Double-crown-retained: Marburger double crowns with TC-SNAP system	125 patients with 125 RDPs (460 abutment teeth): 55 patients with 55 RDPs with rigid support (4 or more abutment teeth with a definite terminal stop between inner and outer crown) and 70 patients with 70 RDPs with resilient support (3 or fewer abutment teeth with mucosal support)	n.a.	n.a.	n.a.
Widbom, 2004 [43]	Retrospective	3.8 years (9 months-9.3 years)	Double-crown-retained with various replaceable snap attachments (Ipso-clips)	72 patients with 75 RDPs	n.a.	One examination after mean observation period of 3.8 years (9 months-9.3 years)	n.a.
Wolfart, 2012 [44]	RCT	5 years	Attachment-retained: precision attachments with removable plastic inserts	81 patients with RDPs with precision attachments	71 patients with fixed partial dentures with shortened dental arch	6 weeks, 6 months and then annually for 5 years after treatment	n.a.
Wöstmann, 2007 [45]	Retrospective	5.3 ± 2.9 years	Double-crown-retained: cylindric	463 patients with 554 parallel-sided cylindric double-crown-retained RDPs (1758 abutment teeth)	n.a.	Once a year	Same patient collective as Rehmann, 2006; different types of RDPs summarized (lack of differentiation)
Yoshino, 2020 [46]	Retrospective	12.7 ± 6.6 years	Double-crown-retained	174 patients with 213 RDPs	n.a.	Regular	No precise description of the kind of double crown
Zierden, 2018 [47]	Retrospective	3.87 ± 3.15 years (nonprecious alloy: 2.99 ± 2.52 years; precious alloy: 5.36 ± 3.53 years)	Double-crown-retained	462 patients with 572 RDPs	n.a.	Once a year	Different types of double crowns: all types summarized for the evaluation (lack of differentiation)

It is important to note that these classifications only refer to the quality of information regarding technical complications and failure reporting but do not judge the quality of the respective publications or clinical studies themselves in any way.

The results of the bias assessment and the quality of the studies according to GRADE as related to the research question at hand are shown in Table 3. Three RCTs achieved a high-quality rating (+ + + +), seven RCTs achieved a moderate-quality rating (+ + +), six retrospective/prospective studies achieved a low-quality rating (+ +), and 27 studies achieved a very low-quality rating (+).

For a more detailed assessment of bias, the Appendix A includes a risk of bias assessment according to the Cochrane library, once for RCTs (RoB 2 tool) and once for non-randomized studies of intervention (ROBINS-I tool) (Appendix A).

## 3. Results

### 3.1. Study Selection

The original electronic search, which was conducted in 2018, returned 12,994 results (Figure 1). Thirty-two publications were included after the final consensus conference of the three independent investigators (O.M., R.L. and H.R.). An updated search was performed regularly and was carried out for the last time in June 2022. The period of time which was considered during this updated search started on 1 January 2014 and ended on 29 June 2022. The electronic search resulted in 6230 hits. After an analysis of the titles, 785 articles were included. The abstract-level analysis then resulted in the inclusion of 150 articles, from which the full texts were obtained. After analysis at the full-text level, duplicates were excluded that were already a part of the original search results (2018). Then, the consensus conference was held for all papers that had received a differing evaluation by two or more of the authors (M-T.D., H.R. and K.K.), and finally, 11 additional full texts were included. All identified papers were available in full text.

All the studies that were excluded did not meet the inclusion/exclusion criteria. The procedure resulted in a total of 43 included full texts (Figure 1).

### 3.2. Study Characteristics

#### 3.2.1. Study Design

Among the included studies, 10 RCTs, 3 prospective studies and 30 retrospective studies were found. There were two multicenter studies. Five studies were written in German and 38 in English.

In all RCTs, blinding was not possible due to the respective therapies.

The mean observation period of the studies varied between two and ten years (Table 2). In a few studies, some subgroups were followed for only six months (Table 2), but because either the mean observation time of all cases was equal to or more than two years or a large proportion of the included patients were followed for a period that was equal to or more than two years, these studies were still included [5,9,25,30,33,42,43].

Treatments predominantly took place in universities but also occurred in military hospitals and were performed by both dentists and students under supervision.

The same patient collective was followed over several years by the author groups of Bergman [10,11,12], Budtz-Jørgensen [14,15] and Stober [34,35]. Different parameters related to the same patient collective were analyzed by Bergman [13] and Ericson [16], Rehmann [27] and Wöstmann [45], Hahnel [17] and Behr [8], and Vanzeveren [38,39].

#### 3.2.2. Participants

All the study participants had a moderately reduced dentition requiring treatment. Only four papers presented exclusion criteria [19,20,21,47], and only two papers reported inclusion criteria [32,46]. Both inclusion and exclusion criteria were listed in just two studies [37,44]. In the study by Kapur et al. [20], reference was made to a previous publication in which inclusion/exclusion criteria were named.

Participant numbers were heterogeneously distributed between studies. If there were unequal group sizes within a study, the study was included if at least one group met the inclusion criteria [30].

#### 3.2.3. Prosthetic Therapy

The study designs were very variable: some publications investigated single prosthesis types with only partly different designs, whereas others compared groups of differently retained prostheses with each other (clasps, double crowns, bars or precision attachments). Some authors compared differently retained prostheses with fixed restorations or the concept of a shortened dental arch dentition (Appendix A).

#### 3.2.4. Analyzed Parameters

The parameters analyzed were publication specific and inconsistent. For this reason, a meta-analysis was not feasible. Consequently, the individual parameters are summarized descriptively below.

The following kinds of technical complications and failures were identified:Retention of the prosthesisRetention loss of the anchor crowns (decementations)Fracture of the framework (including the anchoring elements and their repair)Fracture/repair of the denture teeth or veneeringFracture/repair of the acrylic denture base and/or saddlesThe necessity of relining/rebasingOther technical failures/complications: occlusal grinding and a need for the addition of more prosthetic elements, among others.

Details of the study results per prosthesis type and kind of technical complication and/or failure as listed above are available in the online Appendix A.

### 3.3. Results of Analyzed Parameters

In the following subsections, some special features of the studies or the results are highlighted.

#### 3.3.1. Framework Material

Only very few studies analyzed different framework materials (titanium, a cobalt-chromium alloy/nonprecious metal alloy and a precious metal alloy) in various combinations [6,9,43,47]. In two studies, however, the majority of frameworks were made of the cobalt-chromium alloy, which precludes comparability in terms of complications and failures with other framework materials [9,43]. Although Au et al. [6] describe noticeable differences in early-occurring failure types and maintenance, the authors are in agreement with Zierden et al. [47] and conclude that the framework material has no significant influence on prosthesis survival.

#### 3.3.2. Retention of the Prosthesis

Only four studies [10,28,44,47] identified retention loss as the most common complication (Figure 2). 

Furthermore, these results refer to three different kinds of prostheses: precision attachment-, clasp- and double-crown-retained ones.

In general, retention was assessed by the dentist or the patient in a purely subjective manner. Only in one article [41] was objective data on prosthesis retention collected via an appropriate measurement procedure.

Mock et al. [23] reported for double-crown-retained prostheses with friction fit that retention loss occurred primarily in the mandible and after abutment tooth loss. A significant deterioration in retention after about a five-year observation time was found by Bergmann et al. [13].

However, yet again, only two studies [11,45] described how the complication of retention loss was handled.

A detailed description (laser welding to a secondary crown) was only given by Wöstmann et al. [45]. According to the data of Bergmann et al. [11], activation of the clasps to counteract the loss of retention can be assumed.

#### 3.3.3. Retention Loss of the Anchor Crowns (Decementation)

The retention loss of the anchor crowns was mentioned as the most frequent complication by the second highest number of publications [7,8,23,24,31,43] (Figure 2).

A significant difference between the two anchorage forms (conical and electroplated) could not be shown by Stober et al. [34,35]. Behr et al. [7] reported that decementation was the most common technical complication for both anchorage forms (parallel-sided and conical), and Mock et al. [23] described the same for double-crown-retained prostheses with a friction fit. In addition, a significant difference between the sexes was found: after nine years, the probability that no primary crown had decemented was 73% in women and 45% in men.

The long-term results of Behr et al. [8] showed that after 15 years, a decementation occurred in 75% of patients. Furthermore, the authors investigated different types of double crowns in conjunction with different cements with regard to decementation rates. However, only double crowns cemented with temporary cement (zinc oxide-eugenol cement) showed a higher decementation rate.

Bergmann et al. [13] reported 25 decementations on 13 anchor teeth; therefore, there must have been multiple decementations per tooth. Multiple decementations were also found by Ishida et al. [19] for four abutment teeth (2.7%).

As one of the few groups of authors, Ishida et al. [19] described, in detail, the necessary maintenance as a result of the complications that occurred. Sixteen (11.1%) of the decemented abutment teeth could be maintained with full function by recementation alone, whereas four (2.7%) other abutment teeth needed a post and core treatment.

#### 3.3.4. Fracture of the Framework (Including the Anchorage Elements and Their Repair)

Vanzeveren et al. [39] reported that most clasp fractures (*n* = 20, 2.5%) occurred in short clasps (equipoise and RPI (rest, palatal plate, i-bar) clasps). In this publication, the percentage refers to the number of post-examined abutment teeth. In addition, 22 of the 27 fractures (2.74%) were related to free-end situations (Kennedy class I and II).

As seen for the retention loss of abutment teeth, some fractures also occurred multiple times in the same patient, as described by Widbom et al. [43].

In an observation period of up to ten years, only a few remakes of prostheses were required due to framework or clasp fractures (<5.2% or *n* ≤ 2) [11,12,21,25].

#### 3.3.5. Fracture/Repair of Denture Teeth or Veneering

Regarding the fracture/repair of denture teeth or veneering, the highest number of publications agreed that this is the most frequent complication [15,16,17,26,29,32,34,35,36,41].

One study [29] gave a precise differentiation with regard to the teeth (anterior teeth, premolars and molars) that were affected by veneering fractures. It was found that the premolars most frequently exhibited veneer fractures. The authors did not find a significant difference between the tooth types in terms of fractures of the veneers.

Stober et al. [36] indicated no significant difference between the wear of denture teeth in telescopic and complete dentures. However, the authors found that denture tooth wear was greater if all-ceramic crowns or bridges were present in the opposing jaw.

#### 3.3.6. Fracture/Repair of the Acrylic Denture Base and/or Saddles

Most publications clearly differentiate between “crack” and “fracture”. Other publications, however, only speak generally of “complications” [19], “adjustments” [11,40,47], “minor repair” [12,14] or “repair” [13,27,44,45].

Survival rates are described exclusively by Vermeulen et al. [40]. The tooth-supported prostheses in this study showed a higher survival rate of 75/82% (lower/upper jaw) after five years and 55% (both jaws) after ten years as compared to that of the extension base prostheses of 60/65% (lower/upper jaw) after five years and 40/41% (lower/upper jaw) after ten years.

The number/percentage of new prostheses needed is mentioned only by Yoshino et al. and Kurosaki et al. [22,46]. There were 4.7% (*n* = 10) remakes required [46] in a long-term study that lasted up to 12.7 years. In the study by Kurosaki et al. [22], four cases resulted in a discontinuation of wearing the respective prostheses (no replacement), and another four cases resulted in a change in treatment concept (implant-supported restoration).

#### 3.3.7. Relining/Rebasing

The need for relining/rebasing was mentioned as the most frequent complication by the second highest number of publications [9,20,21,30,38,45] (Figure 2).

Although in the more recent publications, “relining” is used according to its definition and in differentiation to “rebasing” [48], these terms seem to have been possibly used synonymously in older publications [20,21]. Bergmann et al. [11,12] are the only group of authors to describe a remount procedure (“rebasing and occlusal grinding”) in distinction from “relining”.

Schulte and Smith [30] indicated that the need for relining became apparent after an average wear period of 36.9 months.

#### 3.3.8. Other Failures/Complications

Five groups of authors reported complications related to occlusion [6,14,27,44,47]. Au et al. [6] reported either nonocclusion or too high occlusion. A necessary correction due to occlusion that was too high was also reported by Rehmann et al. [27]. Two groups of authors also reported an adjustment or correction of the occlusion [44,47]. Budtz-Jørgensen et al. reported a necessary grinding [14].

Complications associated with a poor fit of the prosthesis were reported in three publications [5,6,25]. The complications were described as either “ill-fitting” [5] or a “poor (clinical) fit (and adaptation)” [6,25].

The need for the remake or repair of primary and/or secondary crowns was described by two groups of authors [45,47].

Additionally, two groups of authors reported complications related to anchoring elements [6,25]. Au et al. [6] reported a “retainer not connecting to tooth”, and Nisser et al. [25] reported a “need for adding more prosthetic elements”.

One publication reported a fracture of the soldering [7] and another the failure of the electroplated structure [32].

## 4. Discussion

The aim of the present publication was to prepare a systematic literature review on technical complications and failures of removable partial dentures in moderately reduced dentition with a subsequent meta-analysis. In spite of a thorough search of the literature over several years and evaluation by different experienced researchers, and the conducting of consensus conferences, it was not possible to extract the required information regarding the technical complications from the literature found. The technical complications have not been reported with sufficient detail and precision in the literature or have been published incompletely or with ambiguous reference values. Based on these findings, a reporting scheme for future publications is suggested.

The heterogeneous data situation (Appendix A) made performing a meta-analysis impossible. Due to this heterogeneity, a graphical representation of the data, e.g., in the form of a forest plot or a similar graph, was not possible. Consequently, the results of the studies were tabulated and analyzed descriptively.

Removable prostheses are subject to great design variability. Although some design elements are common (denture saddles, prefabricated denture teeth and major connectors), others are specific (clasps in clasp-retained RDPs; fabricated or custom attachments in attachment-retained dentures; and telescopic, conical or galvanic double crowns in double-crown-retained dentures). Thus, certain technical complications may occur with all types of prostheses, whereas others are prosthesis-type specific. Against this background, an initial subdivision of the prostheses according to the anchorage type (precision attachment-, clasp- or double-crown-retained) was made for the tabular presentation. Thus, full comparability between prosthesis types is only possible for prostheses with the same construction principle.

Furthermore, very few studies comparing different types of anchorages (and not only different forms of the same type of anchorage) emerged [19,25,33,38,39,40,41]. However, it must be noted that a randomized prospective study comparing all three commonly used anchorage types (clasps, telescopes and attachments) is not feasible for individual patient reasons (clasps in the visible area and grinding of teeth) and for ethical reasons (preparation of native teeth and inclusion of teeth due to the necessary anchorage). Consequently, it is often true that only retrospective studies can be used for this purpose. However, according to GRADE, these studies per se are of low quality (+ +). Further inconsistencies (data missing and merging of study groups with different anchoring types) and the resulting deduction of a quality grade are more significant here than in the case of RCTs, which are of high quality (+ + + +).

The classification of the quality of the studies refers only to the present research question of this systematic review and does not explicitly represent a general statement about the quality of the studies per se. Compared to a previous publication [1] related to biological complications of removable partial dentures, in which a direct reference to the tooth was always given due to the research question, the differently selected reference values (per patient, per prosthesis and per tooth) proved to be very difficult to apply to the present research question. This often resulted in missing data for the GRADE evaluation.

In addition to the classification by prosthesis type, the most frequent reasons for complications/failures were recorded and presented in Appendix A. It should be noted that certain complications/failures are more frequent/less frequent depending on the type of prosthesis. For example, with clasp-retained RPDs, fractures of the clasps are frequent [6,9,19], but decementations are rare [19].

The frameworks were predominantly made of a cobalt-chromium alloy, although some studies also investigated other framework materials [6,9,43,47]. The situation was much more heterogeneous for the anchorage elements, as clasps were usually made of the same alloy and produced in the same casting, whereas attachments or double crowns were connected to the framework by soldering, lasering or bonding [32,34]. Mostly, the exact procedure was not described.

The retention of a removable prosthesis is a parameter that makes an essential difference in regard to fixed restorations, especially for patients, and is determined by the anchoring elements. This was shown in a study by John et al. [49], who showed a 1.9 times higher problem rate determined with the OHIP-G49 questionnaire in patients who were treated with removable/complete dentures as compared to patients who were treated with fixed dentures. Thus, the removable nature of RPDs seems to be a relevant factor for patient satisfaction. The authors [49] concluded that any type of denture—fixed, removable or complete denture—leads to an improvement in quality of life in 96% of the patients studied. This conclusion is consistent with the study results of Vermeulen et al. [40]. Only a few events defined as “failure” could be assigned to a “not wearing” group. The rate after ten years was 4–5% regardless of the anchorage type but slightly higher for clasp-retained RPDs in the mandible (8%).

In addition to the functional limitations due to a loss of retention, psychological consequences could also be conceivable as a result. A study by Koshino et al. [50] investigated not only the quality of life but also the influence of removable dentures on the psyche. It was found, although the results were not statistically significant, that existing complete dentures achieved the lowest quality of life index in relation to psychological factors. The values for new complete dentures in one or both jaws and for RPDs were close to each other and higher than the values for existing complete dentures.

Among the included studies, only Mock et al. [23] mentioned this aspect. Here, the authors described that the subjective perception or acceptance of RPDs may well show a discrepancy with objective findings (loss of retention).

Only in one study [41] could an exact description of retention via a simple clinical measurement method using the withdrawal force in Newton be found. Other studies gave a description of retention based on subjective criteria such as “inadequate” and “good”, etc., that were determined through the self-reporting of dentists or patients. In some studies, only the description of the problem itself was found in terms of a necessity for an increase or decrease in retention. This made a comparison between the studies’ results almost impossible.

In addition to the straightforward method of Wagner et al. [41], other methods for the measurement of prosthesis retention can also be found in the literature. For example, Bayer et al. [51] developed a special measuring device for this purpose.

Depending on the type of anchorage, there are different possibilities for adjusting the retention. For example, in clasp-retained RDPs, the activation or deactivation of the clasps is possible to a limited extent. Bergman et al. [11] described clasp adjustment as the most common form of post-treatment. In the case of attachment-retained RDPs with precision attachments, the replacement of the plastic inserts is possible and ensures the long-term functionality of prefabricated attachment-retained prostheses with a small but regular effort [44]. Increasing the retention in double-crown prostheses is often only possible with considerable technical effort. Of the included studies, only Wöstmann et al. [45] described in detail how retention has been increased. In addition to the method described by Wöstmann et al. [45], there are also other possibilities for increasing the friction of double-crown-retained RDPs. For example, two to four laser dots can be introduced into the inner lumen of the secondary crown, the joint gap can be filled with a layer of plastic or silicone, a layer of electroplated gold can be introduced into the inner lumen of the secondary crown, or a groove can be milled into the primary crown, and a “clip” can be incorporated into the secondary crown after fenestration. For telescopic crowns with an existing mesostructure, this can be remade or replaced [52].

These maintenance or repair measures are only applicable for denture types whose anchorage elements are connected to the residual dentition via fixed dentures. It is not applicable for clasp-retained RDPs whose clasps are retained to healthy teeth, or attachment-retained RDPs whose male attachments (patrices) are adhesively attached to healthy teeth.

Consequently, there was only one study with clasp-retained RDPs that reported the decementation of clasp-bearing crowns [19]. For most studies, it can be assumed that clasps were only applied to healthy or conventional filling-restored teeth.

A comparison between the studies is again difficult, as different reference values were selected for each one: decementation per anchor tooth, per prosthesis, per patient or a mixture thereof. Likewise, the subsequent maintenance or repair measures are rarely described. Therefore, it is not always possible to say whether just a recementation was necessary or whether further measures such as die reconstruction or the insertion of a root post were required.

Fracture reporting often lacks a clear differentiation between major connectors, minor connectors and clasps, as, most commonly, only the “framework fracture” is mentioned.

Although there are indications that fractures tend to occur in the area of the clasps or the minor connectors [39], this cannot be extrapolated. There is also no information on how the fractures were repaired. Furthermore, it mostly remains unclear if there was a distinction between complication and failure. Based on the information given in some studies concerning remakes of prostheses, the inference could be made that the respective fracture was not repairable and, thus, has to be defined as a failure [11,12,21,25].

Fractures of denture teeth and veneering are complications that can affect all denture types. Veneering fractures occur, in particular, with attachment- or double-crown-retained-RDPs, but also with clasp-retained RDPs. The latter occurs when pontic designs or clasp-bearing crowns are used. The fracture of denture teeth or veneering was the most frequent complication, as identified by the majority of publications.

No details on repair measures were found for this type of complication either. However, the measures required to replace or reattach a denture tooth or renew a veneering are less complex or technically challenging than framework repairs for a major connector or retention increase in a double-crown prosthesis.

The fracture of the acrylic base is also a complication that can affect all types of prostheses. This complication may be interlinked with the need for relining, as the latter could be causative under certain circumstances. However, this possible causal link was not investigated or reported in any of the included studies.

In the included studies, there is a diverse description/differentiation of the occurring fractures or necessary repairs. Thus, the results can only be partially compared. Again, the necessary repair measures were not described. However, repairs of this kind of complication can be assumed as being rather simple since only the acrylic part of the prosthesis is involved.

Relining is a recommended aftercare measure for removable dentures. It is intended to restore the fit of the denture saddles after jaw bone recession to ensure better force transmission. Although there are no evidence-based recommendations on frequency, regular intervals are recommended. A study by Pham et al. [53] described a statistically significant influence of relining frequency on bone resorption (posterior residual ridge resorption). In the included studies, however, only Schulte and Smith [30] gave an exact time period (36.9 months) after which relining was necessary. However, it should be borne in mind that the prostheses in the study were of a special design that is no longer used today. There was also no description of the basis on which a need for relining was determined in the included studies.

Authors sometimes refer to rebasing rather than relining. The suspicion is that both terms might have been used synonymously, although they should not be used synonymously according to the glossary of prosthodontic terms [48]. However, due to the lack of a precise description of the measures, this suspicion cannot be confirmed or refuted.

Depending on the respective study, parameters other than those mentioned above were also analyzed. Some of these deserve special attention. A need for occlusal adjustment was described by five authors [6,15,27,44,47]. This is also represented as a comprehensible post-treatment measure. Ill-fitting RPDs were described several times by different authors. Unfortunately, the reasons were not given (e.g., production-related, not wearing the RPDs, dropping the RPDs, etc.).

As far as can be determined, the necessary remakes of primary and secondary parts described by two authors [45,47] did not cause any remakes of the prostheses. In contrast, the complications related to anchoring elements in one [25] of the two studies [6] led to a necessary remake of the prostheses. The other complications were only described in one study (e.g., failure of electroplated structure and soldering fractures) [7,32].

For all other described failures/complications, no exact repair measures were given.

Future studies should follow the following reporting scheme to allow for comparability of the results. We recommend:Clear definition of complication and failure. Everything that is repairable and does not lead to a new prosthesis is a complication. Everything that cannot be repaired and leads to a new prosthesis is a failure.Mandatory specification of absolute and relative frequencies.Avoiding too much differentiation, e.g., the use of the Kennedy subgroup classification or the subdivision of veneering fractures by tooth type. Although these differentiations are helpful for classifying different subgroups, they run up against a hard case limit in terms of study design. Due to the required number of cases per group, such studies would only be feasible in a multicenter setting.Avoiding mixing subgroups of considerably different group sizes. On the one hand, very small case numbers mixed into another group (e.g., different anchoring elements or framework materials) may be confounding factors; on the other hand, biometrical problems arise from very small case numbers regarding comparability.Reporting information on which complication occurred first and then whether or which complications followed.Objectively measuring the retention of prostheses (e.g., with a spring balance) is highly recommended.Indicating the number of decementations per prosthesis, but also in relation to the abutment teeth. Furthermore, it should be indicated which exact restoration parts have been decemented (e.g., only the restoration or also a post and core build-up).Specifying, in the case of fractures, exactly what has been fractured.Describing in detail the necessary repair measures or a necessary new fabrication in relation to all complications/failures.

## 5. Conclusions

In terms of technical complications, different studies provide very different data. For this reason, only a descriptive description could be given. In all analyzed papers, the fracture/repair of the denture teeth or veneering was identified as the most common complication.

In the future, in order to improve the comparability of studies on the complications/failures of removable dentures for moderately to severely reduced dentition, the described reporting scheme should be applied.

## Figures and Tables

**Figure 1 dentistry-11-00055-f001:**
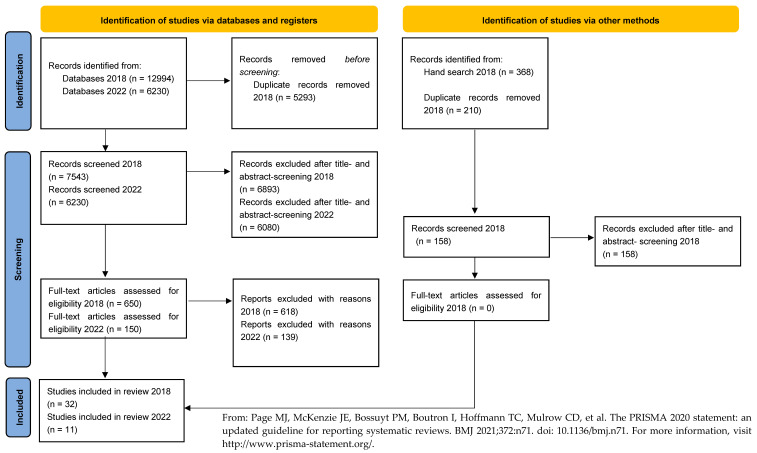
Flowchart of the literature selection [3].

**Figure 2 dentistry-11-00055-f002:**
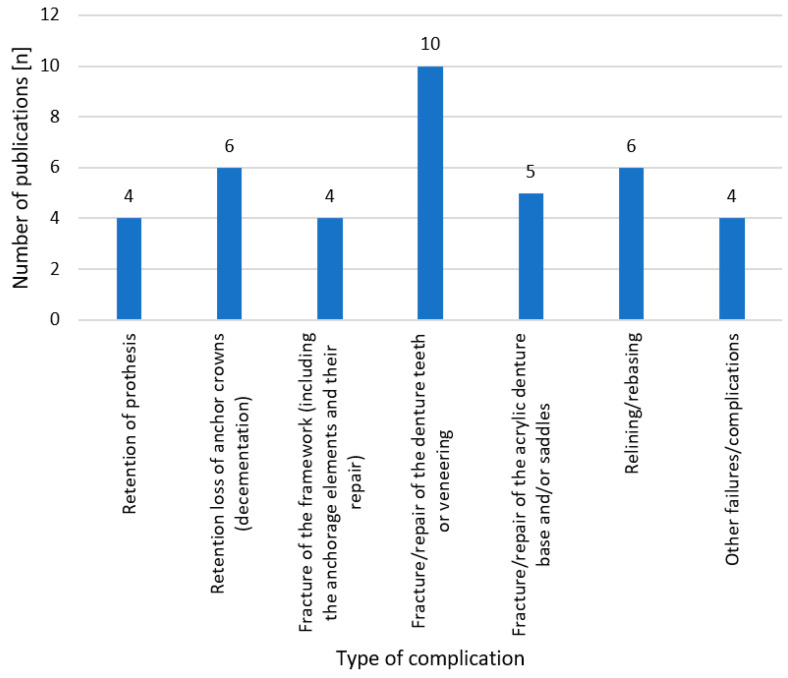
Number of publications per complication type identified as most common in the respective paper. Four studies could not be included because only one result (*n* = 0, 0%) was given [42] or because absolute or relative frequencies were used alternately and with changing reference groups [11,12,13].

**Table 1 dentistry-11-00055-t001:** PICO scheme of the systematic review.

P (population)	Patients with moderately reduced dentition.
I (intervention)	Therapy with some type of removable denture.
C (comparison)	Therapy with another type of removable prosthesis, fixed prosthesis or leaving the situation as is.
O (outcome)	Information on time-related technical failure/complication rates.

**Table 3 dentistry-11-00055-t003:** GRADE quality rating for the included studies.

Author, Year	Risk of Bias/Limitations	Inconsistency	Indirectness	Imprecisions	Publication Bias	Quality
Al-Imam, 2016 [5]	Low	No	No	Data missing	No	+
Au, 2000 [6]	Low	No	No	No	No	+ + + +
Behr, 2000 [7]	Low	No	No	Data missing	No	+
Behr, 2009 [8]	Low	No	No	Data missing	No	+
Behr, 2012 [9]	Low	No	No	No	No	+ +
Bergman, 1971 [10]	Low	No	No	Data missing	No	+
Bergman, 1977 [11]	Low	No	No	Data missing	No	+
Bergman, 1982 [12]	Low	No	No	Data missing	No	+
Bergmann, 1996 [13]	Low	No	No	Data missing	No	+
Budtz-Jørgensen, 1987 [14]	Low	No	No	Data missing	No	+ + +
Budtz-Jørgensen, 1990 [15]	Low	No	No	Data missing	No	+ + +
Ericson, 1990 [16]	Low	No	No	No	No	+ +
Hahnel, 2012 [17]	Low	No	No	Data missing	No	+
Heydecke, 2003 [18]	Moderate	No	No	No	No	+
Ishida, 2017 [19]	Moderate	No	No	No	No	+
Kapur, 1989, Part II [20]	Moderate	No	No	No	No	+ + +
Kapur, 1994, Part I [21]	Moderate	No	No	No	No	+ + +
Kurosaki, 2021 [22]	Low	No	No	Data missing	No	+
Mock, 2005 [23]	Low	No	No	No	No	+ +
Nickenig, 1995 [24]	Low	No	No	Data missing	No	+
Nisser, 2022 [25]	Moderate	No	No	Data missing	No	+
Pihlaja, 2015 [26]	Low	No	No	Data missing	No	+
Rehmann, 2006 [27]	Moderate	No	No	Data missing	No	+
Schmitt, 2011 [28]	Moderate	No	No	No	No	+
Scholz, 2010 [29]	Low	No	No	No	No	+ +
Schulte, 1980 [30]	Low	No	No	Data missing	No	+
Schwindling, 2014 [31]	Moderate	No	No	No	No	+
Schwindling, 2017 [32]	Low	No	No	No	No	+ + + +
Stegelmann, 2012 [33]	Low	No	No	Data missing	No	+
Stober, 2012 [34]	Low	No	No	Data missing	No	+ + +
Stober, 2015 [35]	Low	No	No	Data missing	No	+ + +
Stober, 2020 [36]	Low	No	No	Data missing	No	+
Thomason, 2007 [37]	Low	No	No	Data missing	No	+ + +
Vanzeveren, Part I, 2003 [38]	Moderate	No	No	No	No	+
Vanzeveren, Part II, 2003 [39]	Moderate	No	No	No	No	+
Vermeulen, 1996 [40]	Low	No	No	No	No	+ +
Wagner, 2000 [41]	Moderate	Yes	No	No	No	+
Wenz, 2001 [42]	Low	No	No	Data missing	No	+
Widbom, 2004 [43]	Low	No	No	No	No	+ +
Wolfart, 2012 [44]	Low	No	No	No	No	+ + + +
Wöstmann, 2007 [45]	Moderate	No	No	No	No	+
Yoshino, 2020 [46]	Low	No	No	Data missing	No	+
Zierden, 2018 [47]	Moderate	No	No	No	No	+

## Data Availability

The data presented in this study are available in this systematic review and the Appendix A.

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
