# Peer review of "Technical Complications of Removable Partial Dentures in the Moderately Reduced Dentition: A Systematic Review"

_dentistry, 2023, doi:10.3390/dj11020055_

Round 1

Reviewer 1 Report

Review Dentistry Manuscript 2199174

Technical complications of removable partial dentures in the moderately reduced dentition: A systematic review

The article is interesting and has potential for publication at Dentistry, but, some major points should be considered. These are detailed as follows:

Title – Abstract -- Introduction

Neither the title, abstract nor introduction adequately explains the meaning of the term “moderately reduced dentition”. Authors are encouraged to better explain it. Are they considering Kennedy’s classification? Kennedy’s Class I or II arches? The presence of less than a certain number of teeth? Please clarify it, since this is a key point for the article. Considering that this was one of the PICO terms (population: Patients with moderately reduced dentition), and the authors affirm in the last sentence of the introduction that a “meta-analysis was not performed due to inconsistent parameters, It is important to consider that maybe the systematic search was inconsistent due to the absence of arch appropriate search criteria.

Materials and Methods

This section needs some considerations, as well:

1)     The “search terms” are not present. The authors explain they were the same as 2 previously published articles, but considering that they have supplied a Supplementary File with the Summary of included studies, they could be added there as well. Also, it would be important to better describe: a) the databases used for the search; (Only PubMed? If so, why?); 2) the search strategies (describing the Medical Subject Headings (MeSH) terms used; (3): the time period when searches were performed (this information is available only at the Result section. This is critical to allow adequate reproduction if the results.

2)     Again, I consider the criteria for differentiating a “moderately reduced dentition” essential for the comprehension of the article. The Exclusion criteria say that “Three or fewer residual teeth” were considered a cut-point for the article selection. So, it makes me think that, if 4 or more teeth may characterize a “moderately reduced dentition patient”, and none teeth a totally edentulous one, how many teeth a patient may have to be considered as a “severely reduced dentition” patient? Just 3 to 1 tooth remaining? And what is the maximum number of teeth a patient may possess to not characterize it as a “minimum reduced dentition”? Please explain this.

This is critical to the comprehension and evaluation of the Results and the Discussion section since a Systematic review that does not present an adequate search strategy and PICO question will probably fail in their proposition of comparing the relevant literature.

Results

Results are adequately present, considering the difficulties in comparing the diversity of studies.

Discussion

The section is well-written and informative, but it shows that there is great variability of variables and results, and this is an important factor that may explain the difficulties in performing a meta-analysis, for example. So, the limitations of the present work should be discussed as well, since suggestions for future studies are present on this topic.

Author Response

Point 1:

Title – Abstract – Introduction and Materials and Methods 2)

Neither the title, abstract nor introduction adequately explains the meaning of the term “moderately reduced dentition”. Authors are encouraged to better explain it. Are they considering Kennedy’s classification? Kennedy’s Class I or II arches? The presence of less than a certain number of teeth? Please clarify it, since this is a key point for the article. Considering that this was one of the PICO terms (population: Patients with moderately reduced dentition), and the authors affirm in the last sentence of the introduction that a “meta-analysis was not performed due to inconsistent parameters, It is important to consider that maybe the systematic search was inconsistent due to the absence of arch appropriate search criteria.

2)     Again, I consider the criteria for differentiating a “moderately reduced dentition” essential for the comprehension of the article. The Exclusion criteria say that “Three or fewer residual teeth” were considered a cut-point for the article selection. So, it makes me think that, if 4 or more teeth may characterize a “moderately reduced dentition patient”, and none teeth a totally edentulous one, how many teeth a patient may have to be considered as a “severely reduced dentition” patient? Just 3 to 1 tooth remaining? And what is the maximum number of teeth a patient may possess to not characterize it as a “minimum reduced dentition”? Please explain this.

Response 1:

As stated by reviewer 1 it is correct to assume that “…4 or more teeth […] characterize a “moderately reduced dentition […]”, [… while a] “severely reduced dentition” […implies only] 3 to 1 […] remaining […teeth]. In contrast, a “minimum reduced dentition” cannot be well defined in this context but is rather identified by the absence of removable protheses and the presence of fixed protheses. This was clarified in the manuscript (introduction: page 1 line 34; material and methods: page 2 lines 68-73). 

We are positive that the systematic search was not inconsistent as appropriate search criteria were used. These were added as supplementary material.

Changes 1 (introduction: page 1 line 34; material and methods: page 2 lines 68-73):

…treatment for the moderately (4 or more teeth) to severely (3 to 1 remaining teeth) reduced…

Regardless of the Kennedy class, all removable restorations that did not meet the exclusion criteria were included. This is to say that 4 or more teeth characterize a moderately reduced dentition, while a severely reduced dentition implies only 3 to 1 remaining teeth. Restorations for edentulous patients and all fixed restorations were excluded. Studies that investigated both, removable and fixed prostheses in different study arms were not primarily excluded.

Point 2:

Materials and Methods 1)

1)     The “search terms” are not present. The authors explain they were the same as 2 previously published articles, but considering that they have supplied a Supplementary File with the Summary of included studies, they could be added there as well. Also, it would be important to better describe: a) the databases used for the search; (Only PubMed? If so, why?); 2) the search strategies (describing the Medical Subject Headings (MeSH) terms used; (3): the time period when searches were performed (this information is available only at the Result section. This is critical to allow adequate reproduction if the results.

Response 2:

Information was added and the MeSH-Terms, the search strategy and the data bases as well as the time of the first search and the follow up searches, which were published in “Moldovan, O.; Rudolph, H.; Luthardt, R.G. Biological complications of removable dental prostheses in the moderately reduced dentition: a systematic literature review. Clinical oral investigations 2018, 22, 2439-2461, doi:10.1007/s00784-018-2522-y.

Moldovan, O.; Rudolph, H.; Luthardt, R.G. Clinical performance of removable dental prostheses in the moderately reduced dentition: a systematic literature review. Clinical oral investigations 2016, 20, 1435-1447, doi:10.1007/s00784-016-1873-5.” were added as supplementary material (tables S1-1 to S1-3) in order to simplify the readers access to this information. Also, more detailed information was added to the text (material and methods: page 3 lines 90-100).

Changes 2 (material and methods: page 3 lines 90-100):

The search terms, the search strategy, the databases used, and the time periods were described in two previous publications [1,2].

Over the course of this systematic review, the database search in PubMed was updated on a regular basis and last performed on June, 29th 2022. The original search included additional databases (Table S1-2). The hand search ended on January, 15th 2014. The start dates of the hand search can be found in the supplementary material (Table S1-3) and included all volumes of the respective journals. Since neither additional electronic databases nor the hand search revealed additional results on a full text level compared with the PubMed search, these additional sources were not considered for the update searches. A detailed description of the search process can be found in the supplementary material (Table S1-1 to S1-3).

Point 3:

Discussion

The section is well-written and informative, but it shows that there is great variability of variables and results, and this is an important factor that may explain the difficulties in performing a meta-analysis, for example. So, the limitations of the present work should be discussed as well, since suggestions for future studies are present on this topic.

Response 3:

As explained above the search strategy, the databases and the search time line, applied are not causative for the impossibility to perform a metanalysis. Rather inconsistent reporting is responsible for this fundamental lack of information. Based on these findings and in order to improve the situation, a reporting scheme has been suggested (discussion: page 4 lines 189-195).

Changes 3 (discussion page 4 lines 189-195):

In spite of a thorough search of the literature over several years and evaluation by different experienced researchers and the conducting of consensus conferences, it was not possible to extract the required information regarding the technical complications from the literature found. The technical complications have not been reported with sufficient detail and precision in the literature, or have been published incompletely or with ambiguous reference values. Based on these findings, a reporting scheme for future publications is suggested.

Reviewer 2 Report

Systematic review and meta-analysis have significant importance in evidence based practice. However, many studies failed to be included by several reasons. The authors suggested a reporting scheme that prospective authors should follow to enable future meta-analysis.

Author Response

The authors thank for the positive comment. As all papers were considered or rejected based on the clearly defined inclusion or exclusion criteria, we deduce that reviewer two does not see the need for any changes to the manuscript.

Reviewer 3 Report

I read the manuscript with interest.

However, there has to be caution on interpretation as the risk of bias was not assessed properly. Other tools can rate the level of evidence of the included studies. GRADE is used for the statement recommendations strength, not for classifying each study.

Also, the nomenclature and terminology should be corrected. For example, a partial removable dental prosthesis (PRDP) instead of a removable partial denture (RPD).

Author Response

Point 1:

(x) Moderate English changes required 

Response 1:

MDPIs´ english editing service has been used.

Point 2:

However, there has to be caution on interpretation as the risk of bias was not assessed properly. Other tools can rate the level of evidence of the included studies. GRADE is used for the statement recommendations strength, not for classifying each study.

Response 2:

The initial assessment of bias can be found in the GRADE quality rating for the included studies (Table 2). For a more detailed assessment of bias, the risk of bias assessment according to the Cochrane library was used. Here, as recommended by the Cochrane Library, a differentiation was made between the RCTs (RoB 2 tool) and the non-randomized studies of intervention (ROBINS-I tool). A description was added to the manuscript and a detailed version was added as supplementary material (Table S2-1, S2-2 and S3) (abstract: page 1 lines 18-19; material and methods: page 3 lines 124-126). 

Changes 2 (abstract: page 1 lines 18-19; material and methods: page 3 lines 124-126):

The evidence of the included studies was classified using the GRADE system. The bias risk was determined using the RoB2 tool and the ROBINS-I tool.

For a more detailed assessment of bias, the supplementary material includes a risk of bias assessment according to the Cochrane library, once for RCTs (RoB 2 tool) and once for non-randomized studies of intervention (ROBINS-I tool) (tables S2-1, S-2 und S3).

Point 3:

Also, the nomenclature and terminology should be corrected. For example, a partial removable dental prosthesis (PRDP) instead of a removable partial denture (RPD).

Response 3:

For the nomenclature of the type of prosthesis, we followed the glossary of prosthodontic terms ninth edition (The Glossary of Prosthodontic Terms: Ninth edition. The Journal of prosthetic dentistry 2017, 117, e1-e105, doi:10.1016/j.prosdent.2016.12.001.). Thus, we would prefer not to change this wording.

Round 2

Reviewer 1 Report

The authors have adequately explained the questions formulated and performed the suggested modifications. Considering this fact, the article's quality and capacity to reproduce the results have significantly improved.

Reviewer 3 Report

Please used the term PRDP = partial removable dental prosthesis. instead or RPD = removable partial denture